# Laparoscopic Right Hemihepatectomy after Future Liver Remnant Modulation: A Single Surgeon’s Experience

**DOI:** 10.3390/cancers15102851

**Published:** 2023-05-21

**Authors:** Tijs J. Hoogteijling, Jasper P. Sijberden, John N. Primrose, Victoria Morrison-Jones, Sachin Modi, Giuseppe Zimmitti, Marco Garatti, Claudio Sallemi, Mario Morone, Mohammad Abu Hilal

**Affiliations:** 1Department of Surgery, Poliambulanza Foundation Hospital, 25124 Brescia, Italy; tijs.hoogteijling@poliambulanza.it (T.J.H.);; 2Amsterdam UMC Location University of Amsterdam, Department of Surgery, Meibergdreef 9, 1105 AZ Amsterdam, The Netherlands; 3Cancer Center Amsterdam, Cancer Treatment and Quality of Life, 1081 HV Amsterdam, The Netherlands; 4Department of Surgery, University Hospital Southampton NHS Foundation Trust, Southampton SO16 6YD, UK; 5Department of Interventional Radiology, University Hospital Southampton NHS Foundation Trust, Southampton SO16 6YD, UK; 6Department of Interventional Radiology, Poliambulanza Foundation Hospital, 25124 Brescia, Italy

**Keywords:** liver neoplasms, right hemihepatectomy, laparoscopic liver resection, future liver remnant modulation, treatment outcome

## Abstract

**Simple Summary:**

Laparoscopic right hemihepatectomy (L-RHH) after future liver remnant modulation (FLRM) is considered a technically challenging procedure. This study included consecutive L-RHHs performed by a single surgeon, both with and without prior FLRM. The analysis included 59 patients who underwent L-RHH between October 2007 and March 2023, of which 33 patients received FLRM. L-RHH after FLRM was more technically challenging, as it required longer operative time and Pringle duration. However, there were no significant differences in intraoperative blood loss, conversion rates, or postoperative outcomes such as hospital stay, morbidity rates, and textbook outcome. When performed by experienced laparoscopic hepatobiliary surgeons, L-RHH after FLRM is a safe and feasible procedure.

**Abstract:**

Background: Laparoscopic right hemihepatectomy (L-RHH) is still considered a technically complex procedure, which should only be performed by experienced surgeons in specialized centers. Future liver remnant modulation (FLRM) strategies, including portal vein embolization (PVE), and associating liver partition and portal vein ligation for staged hepatectomy (ALPPS), might increase the surgical difficulty of L-RHH, due to the distortion of hepatic anatomy, periportal inflammation, and fibrosis. Therefore, this study aims to evaluate the safety and feasibility of L-RHH after FLRM, when compared with ex novo L-RHH. Methods: All consecutive right hemihepatectomies performed by a single surgeon in the period between October 2007 and March 2023 were retrospectively analyzed. The patient characteristics and perioperative outcomes of L-RHH after FLRM and ex novo L-RHH were compared. Results: A total of 59 patients were included in the analysis, of whom 33 underwent FLRM. Patients undergoing FLRM prior to L-RHH were most often male (93.9% vs. 42.3%, *p* < 0.001), had an ASA-score >2 (45.5% vs. 9.5%, *p* = 0.006), and underwent a two-stage hepatectomy (45.5% vs. 3.8% *p* < 0.001). L-RHH after FLRM was associated with longer operative time (median 360 vs. 300 min, *p* = 0.008) and Pringle duration (31 vs. 24 min, *p* = 0.011). Intraoperative blood loss, unfavorable intraoperative incidents, and conversion rates were similar in both groups. There were no significant differences in length of hospital stay and 30-day overall and severe morbidity rates. Radical resection margin (R0) and textbook outcome rates were equal. One patient who underwent an extended RHH in the FLRM group deceased within 90 days of surgery, due to post-hepatectomy liver failure. Conclusion: L-RHH after FLRM is more technically complex than L-RHH ex novo, as objectified by longer operative time and Pringle duration. Nevertheless, this procedure appears safe and feasible in experienced hands.

## 1. Introduction

Right hemihepatectomy (RHH) is a complex liver resection that is classified as a major surgical procedure, requiring a high level of technical skill [1,2]. The safety of this procedure has been proven with both the open and the minimally invasive approach, but careful attention should be paid to its specific potential complications [3,4]. Resecting the right hemi liver (segments V–VIII) or, in case of an extended right hemihepatectomy, segments IV (or part of IV) –VIII, is associated to an increased risk of post-hepatectomy liver failure (PHLF), which is the main cause of postoperative mortality after major liver resections [5,6]. Such increased risk is related to the potentially insufficient volume of the future liver remnant (FLR), which, on average, corresponds to 35% and 16% of the total liver volume, following RHH and extended right hepatectomy, respectively [7]. For such reasons, in order to reduce the risk of PHLF, FLR volume should always be determined before a major liver resection is performed, as this is strongly associated with the liver’s functional capacity [8,9,10]. Most experts agree that an FLR volume of 20–25% in noncirrhotic, >30–40% in steatotic and cholestatic livers, and >50% in cirrhotic livers should be pursued [11]. In addition, it is advised to perform a functional assessment through hepatobiliary scintigraphy (HBS) with ^99m^Tc-mebrofenin, indocyanine green retention test at 15 min (ICGR15), or newer imaging techniques, such as dynamic hepatocyte-specific contrast-enhanced MRI (DHCE-MRI) with gadolinium ethoxybenzyl diethylenetriaminepentaacetic acid (Gd-EOB-DTPA) [9,12,13,14,15,16].

When the preoperatively determined FLR volume is insufficient, various strategies of future liver remnant modulation (FLRM) have been developed. They include portal vein embolization (PVE) and associating liver partition and portal vein ligation for staged hepatectomy (ALPPS), which stimulate compensatory hypertrophy of the contralateral liver parenchyma, thereby increasing FLR volume and function [9,17,18,19]. Both PVE (in the setting of single- or two-stage hepatectomy) and ALPPS have increased the pool of patients that are eligible for a liver resection [20,21]. However, due to the FLR hypertrophy, the overall liver anatomy can be distorted, making it harder to recognize anatomical landmarks during RHH. In addition, PVE and ALPPS are associated with periportal inflammation and fibrosis, leading to increased difficulty in hilar dissection [22]. 

Another technical breakthrough of the last decades has been the development of laparoscopic liver surgery (LLS). The implementation of LLS was initially slow, but the first consensus statement in Louisville generated enormous enthusiasm for this novel technique [23]. Now, LLS is considered the standard of care in minor liver resections, and is being increasingly used for technically and anatomically major liver resections [1]. In addition, the indications for LLS kept widening, which enabled surgeons in specialized centers to adopt laparoscopy for increasingly difficult resections [24,25,26,27]. Since then, there have been several studies showing favorable outcomes of laparoscopic right hemihepatectomy (L-RHH), and, finally, the Southampton guidelines stated that L-RHH should be expanded further in specialized centers [28,29,30,31,32]. However, studies investigating the results of L-RHH after FLRM are scarce. In this study, we aim to assess the safety and feasibility of L-RHH after FLRM, when compared with L-RHH not preceded by FLRM (ex novo). 

## 2. Methods

This is a retrospective analysis of the prospectively maintained databases of two tertiary referral hepatobiliary centers. All consecutive laparoscopic right or extended right hemihepatectomies performed by a single surgeon (MAH) in the period between October 2007 and March 2023 were included. Patients were stratified in two study groups: those who did and those who did not undergo preoperative FLRM. Baseline characteristics and perioperative outcomes of patients in the two study groups were compared. 

### 2.1. Definitions and Outcomes

The term ‘future liver remnant modulation’ (FLRM) was used to describe either a PVE or a first stage of ALPPS prior to RHH. Resections that were not preceded by FLRM were labeled as ‘ex novo’. Data were collected from electronic health records. Baseline characteristics included patient demographics, body mass index (BMI), American Society of Anesthesiologists (ASA) score, presence of, and, if present, grade (Child-Pugh) of cirrhosis, history of hepatic surgery, neoadjuvant chemotherapy, disease characteristics (pathology, number, and size of lesions), and operative information (type of hemihepatectomy and multiple resections). The Brisbane 2000 terminology was used to define the type and extent of RHH, defining RHH as resection of the right hemi liver (segment 5, 6, 7, 8) and extended right hemihepatectomy (ERHH) as resection of the right hemi liver plus left medial section (segment 4, 5, 6, 7, 8) [33]. Perioperative outcomes consisted of resection margin status, application of Pringle maneuver and Pringle-duration, operative time, intraoperative blood loss, intraoperative transfusion, intraoperative incidents, conversion to an open procedure, length of hospital stay, 30-day morbidity, post hepatectomy liver failure, 30-day readmission, 30-day reintervention, and 90-day or in-hospital mortality. Intraoperative incidents and postoperative morbidity were respectively graded according to the Oslo and the Clavien-Dindo classifications [34,35]. In addition, the rate of textbook outcome in liver surgery (TOLS) was evaluated. TOLS was defined as: the absence of intraoperative incidents of grade 2 or higher, postoperative bile leak grade B or C, severe postoperative complications, readmission within 30 days after discharge, in-hospital mortality, and the presence of an R0 resection margin (in case of malignancy) [36]. 

### 2.2. Technique

A number of publications by our group have detailed the radiological and surgical techniques employed in L-RHH ex novo and following FLRM [3,4,22]. Concisely summarized, the techniques are as follows.

#### 2.2.1. Patient Selection

Patients requiring a liver resection for any indication are discussed in a multidisciplinary team (MDT) meeting with (hepatobiliary) surgeons, pathologists, oncologists, hepatologists, and (interventional) radiologists. In our center, we maintain the following cut-off values for FLR volume: more than 30% in normal background livers, 35% following neoadjuvant chemotherapy, and 40% in the case of underlying chronic liver disease or portal hypertension. HBS is not implemented in our practice, however, in selected cases, ICGR15 tests are performed to assess the functional status of the FLR. All patients needing FLRM first received PVE, and, only if a sufficient FLR hypertrophy was not obtained, a salvage ALPPS was considered. 

#### 2.2.2. PVE

PVE is performed via a trans-hepatic percutaneous ipsilateral approach. Patients are typically treated under sedation and local anesthesia. Under ultrasound guidance, a peripheral vein from the right portal branch is punctured and a vascular sheath is introduced. Portal venography is performed prior to the actual selective embolization. FLR hypertrophy is evaluated by a CT scan 4 weeks after PVE. 

#### 2.2.3. Laparoscopic Right Hemihepatectomy Surgical Technique

Port placement for L-RHH is shown in Figure 1: in order to be in line with the transection plane, which is moved more right due to the left liver hypertrophy, in the group of FLRM, all the ports are usually placed 2 cm right, compared to the ex novo L-RHH group. After accessing the abdominal cavity, a thorough intraoperative ultrasound (IOUS) is performed. The liver is mobilized by first dissecting the round and falciform ligament back to the hepatocaval confluence. Thereafter, the right triangular and coronary ligaments are divided. As the right liver is lifted up and rotated to the left, the inferior retrohepatic vena cava is exposed, and eventual accessory hepatic veins can be identified, dissected, clipped, and divided. The Makuuchi ligament is dissected, slinged, and stapled using a powered vascular stapler (PVS) (ECHELON FLEX™, Ethicon, Johnson & Johnson, New Brunswick, NJ, USA) and the right hepatic vein is isolated and encircled with an elastic tape. After the right liver has been mobilized, the hepatic pedicle is encircled with a cotton tape for Pringle maneuver and the hepatic hilum is dissected, in order to identify the right portal vein (RPV) and the right hepatic artery (RHA), which are dissected and slinged. It should be noted that, after FLRM, this step can be more challenging, due to distorted anatomy and periportal fibrosis [22,37,38]. Typically, the Pringle maneuver is performed by placing a nylon tape around the porta hepatis from the most laterally placed trocar. To facilitate this maneuver, the liver is retracted to the left, thereby putting tension on the hepatoduodenal ligament and exposing the foramen of Winslow. Alternatively, when passing the foramen of Winslow is not possible, due to the earlier mentioned fibrosis, the Pringle maneuver can be applied from the left side of the porta hepatis. To facilitate this more difficult approach, the Goldfinger (Blunt Dissector and Suture Retrieval System, Ethicon Endo Surgery, Johnson & Johnson, New Brunswick, NJ, USA) can be used.

Accurate preoperative CT imaging must be performed in order to better understand the vascular anatomy. By placing a bulldog on the RHA and RPV, the ischemia line on the liver surface between the right and left liver can be identified. In addition, with IOUS and color Doppler, we check the presence of venous and arterial flow in the left, and absence of venous and arterial flow in the right, lobe. Moreover, in the last five years, indocyanine green (ICG) is administered at a dose of 0.3 mg/kg to confirm the negative staining in the right lobe and to ensure the rightness of the ischemic Glissonean line. The RHA is then transected between Hem-o-Lock clips (Weck Closure Systems, Research Triangle Park, NC, USA). The RPV can be divided in a similar fashion, or using a PVS. After FLRM, RPV transection is not necessary, as it has already been embolized or ligated. 

After reassessing the intraparenchymal anatomy with IOUS, the parenchymal transection phase starts. In our center, we use an ultrasonic dissector to transect the Glissonean sheath and the superficial part of the liver parenchyma, and the Cavitron Ultrasonic Surgical Aspirator (CUSA) (Integra Lifesciences, Princeton, NJ, USA) for the deep parenchyma dissection. Titanium or Hem-o-Lock clips are used to control small-medium Glissonean and venous branches. Major vessels, such as the RHD, RPV, and right hepatic vein (RHV), are dissected intraparenchymally and usually transected with a stapler. During the parenchymal transection phase, a good vision of the transection plane is paramount. We use a 30° camera and assert traction on the two sides of the parenchyma to open the liver and maintain a field of vision that is in line with the transection plane. It should be noted that parenchymal dissection can be challenging, due to the presence of embolic material and related inflammation; hence, special attention should be paid to possible stapler failure. 

#### 2.2.4. Salvage ALPPS Surgical Technique 

Ports are positioned as in L-RHH and intra operative ultrasound is regularly performed. No liver mobilization is performed. 

The first step is the identification of the right hepatic artery (RHA), which is slinged and controlled with a bulldog clamp. At this stage, due to the previously performed right portal vein embolization, a clear ischemia line is identified. The right liver ischemia and the adequate arterial and portal flow in the FLR are further confirmed, using intraoperative ultrasound with color Doppler and ICG test, as described previously. The transection line is marked, except in the case of lesion extension in segment IV, in which case the line is deviated further to the left. Thereafter, parenchymal transection is performed with a similar technique to the one described above. The parenchymal transection is extended deep enough to ensure that all major communicating outflow and venous structures are divided (mini ALPPS). After careful assessment of the resection margin, a drain between the resection planes is placed and a PDS-1 10 cm loop is left around the RHA. 

### 2.3. Statistical Analysis

Categorical variables, reported as counts and percentages, were compared between the treatment groups (FLRM and ex novo) using Chi-squared or Fisher’s exact tests, when appropriate. Normally distributed continuous variables were reported as the mean with its standard deviation and compared between treatment groups using an unpaired T-test. Not normally distributed continuous variables were reported as the median with its range and compared between treatment groups using the Mann-Whitney U test. Normality was assessed by visually inspecting histograms and Q-Q plots. A two-sided *p*-value <  0.05 was considered statistically significant. As exploratory analysis, due to the small sample size, unadjusted (univariate) and adjusted (multivariate) regression analyses were performed for the endpoints TOLS, Pringle duration, and operative time. Logistic and linear regression was performed for binary and continuous outcomes, respectively. Besides the exposure (FLRM), potential confounding factors were added as covariates in the adjusted regression analyses when they were significantly (Cut-off: *p* ≤ 0.20) associated with the outcome of interest in the unadjusted analyses. Data were analyzed using R for Mac OS X version 4.2.1 (R Foundation for Statistical Computing, Vienna, Austria).

## 3. Results

Overall, 59 patients that underwent a laparoscopic right or extended right hemihepatectomy were included. As shown in Figure 2, 33 patients underwent L-RHH after FLRM, and 26 patients underwent an ex novo L-RHH. Among the included patients, 45 were operated on in the University Hospital of Southampton, United Kingdom, between October 2007 and October 2019. In November 2019, the operating surgeon (MAH) moved to Fondazione Poliambulanza Hospital in Brescia, Italy, where the remaining 14 patients were treated. In the FLRM group, 28 patients received PVE and five underwent first-stage ALPPS prior to L-RHH.

### 3.1. Baseline, Procedural, and Disease Characteristics

The baseline, procedural, and disease characteristics are shown in Table 1. Both groups were well balanced, in terms of median age, median BMI, and median tumor size. The proportion of male patients and patients with higher ASA scores was significantly higher in the FLRM group than in the ex novo group (93.9% vs. 42.3%, *p* < 0.001, and 45.5% vs. 9.5%, *p* = 0.006, respectively). Significantly more patients in the FLRM group underwent surgery in the setting of a two-stage hepatectomy (45.5% vs. 3.8% *p* < 0.001). The majority of patients (*n* = 32) were treated for colorectal liver metastases (CRLM), whilst 16 were treated for hepatocellular carcinoma (HCC), one was treated for intrahepatic cholangiocarcinoma (CCA), three were treated for non-colorectal liver metastases (NCRLM), and six were treated for benign lesions. The proportion of malignancy was well balanced between the two groups. In the FLRM group, two patients underwent extended right hemihepatectomy. 

### 3.2. Intra- and Postoperative Outcomes 

Intra- and postoperative outcomes are shown in Table 2. FLRM was associated with longer operative times (Median 360 [IQR 300–427.50] vs. 300 min [IQR 240–360], *p* = 0.008) and a longer Pringle duration (Median 31 [IQR 25–43] vs. 24 min [IQR 20–30], *p* = 0.011). The rate of unfavorable intraoperative incidents, conversion to open surgery, and amount of intraoperative blood loss were similar in both groups. The R0 rates did not significantly differ between the FLRM and ex novo groups (90.3% vs. 90%, *p* = 0.970). Postoperatively, the median length of stay, 30-day morbidity, 30-day readmission, and 30-day reintervention rates were comparable and did not statistically differ. The TOLS rates were comparable in the FLRM and ex novo groups (74.2% vs. 70.6%, *p* = 0.788). One patient who underwent an extended L-RHH after FLRM deceased as a result of ISGLS grade B PHLF.

The additional, exploratory, adjusted analyses (Appendix A) confirmed these findings, although the observed difference in operative time no longer reached statistical significance.

## 4. Discussion

The present study reports on the 17-year experience of a single surgeon in performing L-RHH, ex novo and after FLRM [39]. It shows that the added difficulty of anatomical and structural changes after FLRM in L-RHH resulted in a significantly longer operative time and pringle duration. However, importantly, there were no significant differences in intraoperative blood loss, unfavorable incidents, conversion rate and resection margin status between L-RHH after FLRM and ex novo. Importantly, these resections were performed in tertiary referral centers specialized in LLS, and by a surgeon with extensive experience in both laparoscopic and open liver surgery.

L-RHH after FLRM remains a technical challenge. The Southampton guidelines, which stated that the implementation of LLS should be realized in a stepwise manner, owing to the extensive learning curve associated with the more difficult resections, considers L-RHH among the most challenging resections and recommends exploring the technique only in highly specialized centers [30]. FLRM further increases the technical difficulty of L-RHH. FLRM is usually performed in patients with extensive uni- or bilobar disease, who have a high disease burden, and is often used in the context of staged hepatectomy: in this case, the FLR is cleared of lesions during the first stage, followed by PVE and by RHH or extended right hepatectomy during the second stage. Alternatively, during the first stage of ALPPS, the liver parenchyma is (partially) transected before ligation or embolization of the portal vein. A history of previous liver surgery is a well-known factor of increased surgical difficulty [2,40,41]. More importantly, FLR modulation with PVE or ALPPS leads to an important anatomical distortion and periportal fibrosis, which significantly increase the technical difficulty of L-RHH [9]. As a result, evidence regarding the safety and feasibility of these procedures is limited [42].

In this single surgeon experience study, we included all consecutive L-RHH, including two-stage hepatectomies and second-stage ALPPS procedures. Patients in the FLRM group, predictably, were associated with more extensive disease. In addition, FLRM is often performed in the setting of staged hepatectomy, which can be seen from the higher portion of staged hepatectomies in the FLRM group. It is fair to assume that these factors have resulted in the longer operative time and Pringle duration in the FLRM group, even if they did not result in significantly worse intra- and postoperative outcomes. In the current experience, blood loss, transfusion, and conversion rates are largely consistent with previously published series. A study by Fuks et al. (2015), of 26 patients undergoing second-stage L-RHH for CRLM, reported a median blood loss of 250 mL, transfusion rate of 15%, a conversion rate of 15%, a major morbidity rate of 27%, and 9 days length of hospital stay [37]. Another study by Okumura et al. (2019), of 38 patients undergoing second-stage L-RHH for CRLM, reported a median blood loss of 225 mL, 13% transfusion rate, 11% conversion rate, 18% major morbidity rate, and 9 days length of stay [43]. It should be noted that, in the latter analysis, 13 of the 38 patients did not undergo FLRM prior to the second-stage hepatectomy. The most recent analysis was published by Taillieu et al. (2022), reporting the outcomes of seven patients who underwent L-RHH after FLRM. This analysis showed a median intraoperative blood loss of 240 mL, 0% transfusion rate, 1 (14%) conversion, 18% major morbidity, and 4 days hospital stay [44]. In addition, the increased surgical difficulty did not negatively impact oncological efficiency, with an R0 rate approaching 90%, which is comparable to both the ex novo group, and to the reports by Fuks et al., Okumara et al., and Taillieu et al. [37,43,44].

Recently, textbook outcome has been introduced in different surgical disciplines [45]. Based on an all-or-nothing principle, these composite outcome measures incorporate multiple clinical and pathological outcomes to give a more comprehensive picture of patient-level hospital performance [36]. Textbook outcome in liver surgery (TOLS) was defined by Gorgec et al. through a survey among the members of the European African Hepato-Pancreato-Biliary Association (E-AHPBA) and the International Hepato-Pancreato-Biliary Association (I-HPBA), and was subsequently validated in a large retrospective database [36]. As patients undergoing FLRM were excluded from the analyses by Gorgec et al., TOLS is not validated for the patients in the present study. Interestingly, however, when we did apply TOLS to our analysis, the rates in both groups were around 70%, which is consistent with the results reported by Gorgec et al. in the general population of patients undergoing liver surgery [36].

This study has several limitations. First, the small sample size and retrospective nature of the analysis produce an inevitable risk of bias. The baseline characteristics between the two groups differ significantly in terms of gender, history of previous liver surgery, and number of patients with multiple lesions. However, this is one of the largest series reported to date, and we do believe that some baseline differences are inevitable when comparing these two clinically different groups; hence, we chose not to address this by means of statistical techniques such as propensity-score matching. The exploratory, adjusted analyses confirmed that FLRM is associated with a longer Pringle duration, but non-inferior TOLS rates. In these analyses, the FLRM group also tended to have longer operative times, although this finding no longer reached statistical significance. However, the results of these analyses need to be interpreted with extreme caution, due to the very small sample size, making regression analyses notoriously unreliable.

Another issue is the relatively large proportion (56%) of patients who underwent FLRM prior to L-RHH, which is higher than other series [28]. In our center, an effort is made to be as parenchyma-sparing as possible. RHH is typically reserved for patients who have extensive disease, characterized by a large number of lesions or lesions located deep in the parenchyma and in close proximity to major Glissonean or venous vessels. In such scenarios, performing FLRM is often necessary.

Another limitation is that the current results refer to a single surgeon with a wide experience in open and minimally invasive liver surgery, who gradually expanded the indications during the years, whilst accumulating more experience in MILS, thus these results should not be seen as a green light for the liberal adoption of the minimally invasive approach for such complex cases, unless experience is developed and a learning curve has been completed. However, a large single-surgeon series reduces potential confounding factors, thus permitting a more reliable analysis.

## 5. Conclusions

The results of this analysis suggest that, despite the increased technical difficulty, L-RHH after FLRM is feasible and safe for carefully selected patients, assuming that such complex procedures are performed by surgeons highly experienced in LLS.

## Figures and Tables

**Figure 1 cancers-15-02851-f001:**
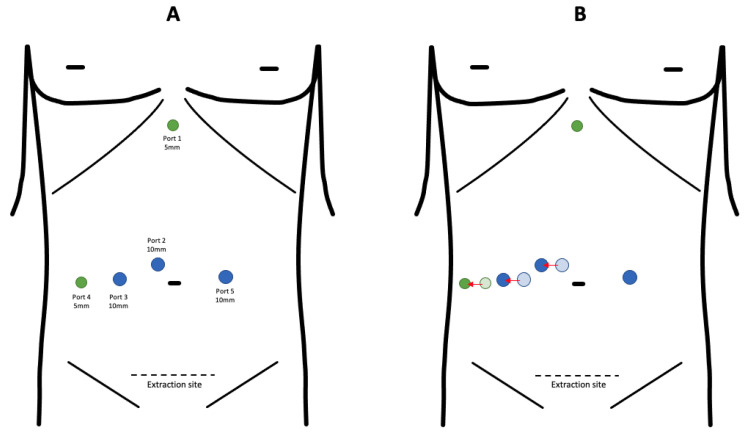
Port placement in L-RHH (**A**) and in L-RHH after FLRM (**B**).

**Figure 2 cancers-15-02851-f002:**
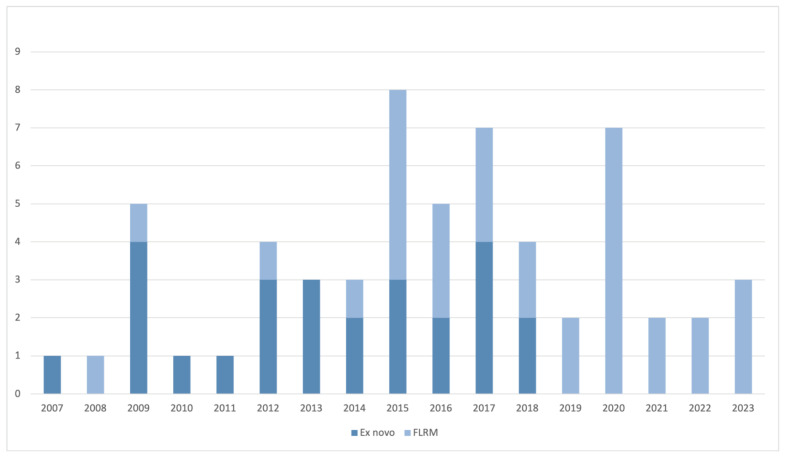
Proportion of L-RHH after FLRM and ex novo per year.

**Table 1 cancers-15-02851-t001:** Baseline Characteristics.

	FLRM (*n* = 33)	Ex Novo (*n* = 26)	*p*
AgeatOp (median [IQR])	64.00 [58.00, 70.00]	60.63 [48.22, 75.42]	0.306
BMI (median [IQR])	28.40 [25.00, 29.70]	27.07 [24.95, 31.00]	0.942
Male Gender (%)	31 (93.9)	11 (42.3)	<0.001
ASA > 2 (%)	15 (45.5)	2 (9.5)	0.006
Cirrhosis (%)			0.036
No	25 (83.3)	24 (100.0)	
Yes *	5 (16.7)	0 (0.0)	
Neoadjuvant Chemotherapy (%)	21 (63.6)	12 (46.2)	0.179
Previous Liver Surgery (%)	15 (46.9)	2 (7.7)	0.001
Pathology (%)			0.040
CRLM	20 (60.6)	12 (48.0)	
HCC	11 (33.3)	5 (20.0)	
CCA	1 (3.0)	0 (0.0)	
Benign	1 (3.0)	5 (20.0)	
NCRLM	0 (0.0)	3 (12.0)	
Malignancy (%)	32 (97.0)	22 (84.6)	0.091
Size Largest Lesion, mm (median [IQR])	40.00 [15.00, 70.00]	44.50 [31.50, 68.50]	0.571
Number of Lesions (median [IQR])	2.00 [1.00, 4.00]	1.00 [1.00, 2.00]	0.124
Type of FLRM (%)			
None	0 (0.0)	26 (100.0)	<0.001
PVE	28 (84.8)	0 (0.0)	
ALPPS	5 (15.2)	0 (0.0)	
Time Interval PVE to Surgery (median [IQR])	43.00 [39.00, 61.00]	-	-
Two-stage Hepatectomy (%)	15 (45.5)	1 (3.8)	<0.001
Extended Right Hemihepatectomy (%)	2 (6.1)	0 (0.0)	0.202
Multiple Resections (%)	4 (12.1)	3 (11.5)	0.945

Abbreviations: FLRM = Future Liver Remnant Modulation; IQR = Inter Quartile Range; BMI = Body Mass Index; ASA = American Society of Anesthesiologists; CRLM = Colorectal Liver Metastasis; HCC = Hepatocellular Carcinoma; CCA = Cholangiocarcinoma; NCRLM = Non-colorectal Liver Metastasis; PVE = Portal Vein Embolization; ALPPS = Associating Liver Partition and Portal vein ligation for Staged hepatectomy. * All patients had Child-Pugh A cirrhosis.

**Table 2 cancers-15-02851-t002:** Perioperative Outcomes.

	FLRM (*n* = 33)	Ex Novo (*n* = 26)	*p*
Pringle Maneuver (%)	30 (93.8)	20 (76.9)	0.065
Pringle Duration (median [IQR]) ^†^	31.00 [25.00, 43.00]	24.00 [20.00, 30.00]	0.011
Operating Time, minutes (median [IQR])	360.00 [300.00, 427.50]	300.00 [240.00, 360.00]	0.008
Intraoperative Blood Loss, mL (median [IQR])	700.00 [400.00, 1200.00]	500.00 [312.50, 737.50]	0.162
Intraoperative Blood Transfusion (%)	9 (28.1)	3 (13.0)	0.182
Intraoperative Incidents, OSLO-classification (%)			0.678
0	26 (83.9)	18 (75.0)	
1	2 (6.5)	3 (12.5)	
2	3 (9.7)	3 (12.5)	
Conversion (%)	2 (6.1)	3 (11.5)	0.453
Length of Hospital Stay, days (median [IQR])	6.00 [5.00, 8.25]	6.00 [5.00, 8.00]	0.537
30-day Complication (%)	8 (24.2)	11 (47.8)	0.067
Severe Postoperative Complications (%)	5 (15.2)	1 (4.2)	0.182
Post-hepatectomy Liver Failure (%)	2 (6.5) ^‡^	0 (0.0)	0.331
30-day Readmission (%)	6 (18.8)	2 (7.7)	0.225
30-day Reintervention (%)	3 (9.4)	1 (5.0)	0.565
90-day Mortality (%)	1 (3.0)	0 (0.0)	0.371
R0 Resection Margin(%)	28 (90.3)	18 (90.0)	0.970
TOLS (%)	23 (74.2)	12 (70.6)	0.788

Abbreviations: FLRM = Future Liver Remnant Modulation; IQR = Inter Quartile Range; OSLO = Oslo classification of intraoperative incidents; ISGLS = International Study Group on Liver Surgery; PHLF = Post Hepatectomy Liver Failure; TOLS = Textbook Outcome in Liver Surgery. Missing data: counts may not add up, due to missing data. Missing values for continuous data: BMI 26; Size of largest lesion 3; Pringle duration 4; Operative time 2; intraoperative blood loss 2; length of stay 2. † Analysis of Pringle duration, only when Pringle maneuver was applied. ‡ One patient had ISGLS grade A, and one had grade B PHLF.

## Data Availability

The data that support the findings of this study are available from the corresponding author, Mohammed Abu Hilal, upon reasonable request. The data are not publicly available, since this could compromise the privacy of research participants.

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
