# Peer review of "Laparoscopic Right Hemihepatectomy after Future Liver Remnant Modulation: A Single Surgeon’s Experience"

_cancers, 2023, doi:10.3390/cancers15102851_

Round 1
Reviewer 1 Report
Dear Editor.
Thank you so much for the opportunity review this interest paper. The authors conducted a retrospetivy study comparing surgical outcome of patients undervwent to laparoscopic right hemihepatectomy (L-RHH) after Future Liver Remnant (FLRM) versus ex novo. All consecutive hemi hepatectomies performed by a single surgeon in the period between October 2009 and September 2022 were analyzed. A total of 58 patients were included in the analysis, of 27 whom 31 underwent FLRM. L-RHH after FLRM was associated with longer operative time and Pringle duration. On the other hand there were no significant differences in length of hospital stay and 30-day overall and severe morbidity rates. I aggre with authors that L-RHH after FLRM is more technically complex than L-RHH ex novo. The paper is well written and may be accepted in present form.
Reviewer 2 Report
I like this paper and would recommend it for publication. The authors present the results of the technically demanding and challenging surgical procedure - the laparoscopic right hemihepatectomy in two groups of patients: 1. after FLR (future liver remnant) modulation and 2. without modulation as a retrospective analysis. The authors proved the feasibility and safety of the mentioned procedure in the experienced hands. This is the case in this paper - the operating surgeon is expert in the field, one of the most experienced liver laparoscopic surgeons in the world. Besides, it is one of the largest series, which was recently published. Last but not least, in my humble opinion, the title of the paper is not quite clear. The group of patients without modulation of the FLR (future liver remnant) is called "ex novo". This expression seems to me a little bit confusing, especially when you read the title for the first time. Perhaps it would be possible the change or modify the title.
Generally, I would like to congratulate the authors on this study.
Reviewer 3 Report
Thank you for providing me with the opportunity to review the manuscript "Laparoscopic Right Hemihepatectomy after Future Liver Remnant Modulation versus ex novo A Single Surgeon's Experience". As a liver surgeon, I found the topic of this study to be very interesting and extremely important. The valuable results presented by surgeons highly experienced in LLS make this paper an important contribution to the field of liver surgery. However, I have provided some comments and suggestions for revision below.
# The results currently presented are based on unadjusted comparisons. To emphasize causal relationships more effectively, it would be advisable to consider adjusted comparisons by controlling for background factors, such as through logistic regression analysis.
# From 2009 to 2022 is a relatively long period, and I am interested in whether there is a difference in the distribution of the implementation of L-RHH after FLRM and ex novo L-RHH.
# I feel that the section with the detailed explanation of the operation could be shortened a bit
# It is better to unify the wording. For example, I see “right hemihepatectomy” and “right hemi hepatectomies”
Minor revisions should be needed.
Round 2
Reviewer 3 Report
Thank you so much for the author's work. I am delighted to endorse a publication.